# Development and Validation of a UHPLC–MS/MS-Based Method to Quantify Cenobamate in Human Plasma Samples

**DOI:** 10.3390/molecules27217325

**Published:** 2022-10-28

**Authors:** Bruno Charlier, Albino Coglianese, Francesca Felicia Operto, Giangennaro Coppola, Ugo de Grazia, Pierantonio Menna, Amelia Filippelli, Fabrizio Dal Piaz, Viviana Izzo

**Affiliations:** 1Department of Medicine, Surgery and Dentistry “Scuola Medica Salernitana”, University of Salerno, Baronissi, 84081 Salerno, Italy; 2University Hospital “San Giovanni di Dio e Ruggi d’Aragona”, 84131 Salerno, Italy; 3Graduate School in Clinical Pathology and Clinical Biochemistry, University of Salerno, Baronissi, 84081 Salerno, Italy; 4Laboratory of Neurological Biochemistry and Neuropharmacology, Fondazione IRCCS “Istituto Neurologico Carlo Besta”, 20133 Milano, Italy; 5Department of Science and Technology for Humans and the Environment, Università Campus Bio-Medico di Roma, 00128 Roma, Italy; 6Operative Research Unit of Clinical Pharmacology, Fondazione Policlinico Universitario Campus Bio-Medico, 00128 Roma, Italy

**Keywords:** antiseizure medications, UHPLC–MS/MS, liquid chromatography, mass spectrometry, cenobamate, therapeutic drug monitoring

## Abstract

Cenobamate (CNB) is the newest antiseizure medication (ASM) approved by the FDA in 2019 to reduce uncontrolled partial-onset seizures in adult patients. Marketed as Xcopri in the USA or Ontozry in the EU (tablets), its mechanism of action has not been fully understood yet; however, it is known that it inhibits voltage-gated sodium channels and positively modulates the aminobutyric acid (GABA) ion channel. CNB shows 88% of oral bioavailability and is responsible for modifying the plasma concentrations of other co-administered ASMs, such as lamotrigine, carbamazepine, phenytoin, phenobarbital and the active metabolite of clobazam. It also interferes with CYP2B6 and CYP3A substrates. Nowadays, few methods are reported in the literature to quantify CNB in human plasma. The aim of this study was to develop and validate, according to the most recent guidelines, an analytical method using ultra-high-performance liquid chromatography coupled with tandem mass spectrometry (UHPLC–MS/MS) to evaluate CNB dosage in plasma samples. Furthermore, we provided a preliminary clinical application of our methodology by evaluating the pharmacokinetic parameters of CNB in two non-adult patients. Plasma levels were monitored for two months. Preliminary data showed a linear increase in plasma CNB concentrations, in both patients, in agreement with the increase in CNB dosage. A seizure-free state was reported for both patients at the dose of 150 mg per day.

## 1. Introduction

Antiseizure medications (ASMs) are the primary pharmacological choice for patients affected by epilepsy, a widely diffused neurological disorder characterized by a heterogeneous range of syndromes, which often demands lifelong treatment, including medication changes and polytherapy [1,2,3]. The aim of these pharmacological treatments is to achieve seizure freedom (100% seizure reduction), avoiding or limiting adverse effects [4]. To date, first-, second- and third-generation drugs are available; the latter demonstrated favorable pharmacokinetics with better bioavailability and fewer drug–drug interactions compared with older drugs [5]. However, despite the development of more than a dozen new molecules over the past 20 years, approximately 30% to 56% of patients with epilepsy continue experiencing uncontrolled seizures during therapy with ASMs [6,7]. 

Cenobamate ([(1R)-1-(2-chlorophenyl)-2-(tetrazol-2-yl)ethyl] carbamate) (Figure 1) is a new antiseizure drug developed by SK Biopharmaceuticals [8], which was approved in 2019 by the Food and Drug Administration (FDA) to treat uncontrolled partial-onset seizures in adult patients, and in March 2021 by the European Medicines Agency (EMA) for the adjunctive treatment of focal-onset seizures in adult patients who have not been satisfactorily controlled despite a previous treatment with at least two ASMs [9]. 

Cenobamate seems quite unique among alkyl-carbamates, which include other ASMs such as meprobamate, flupirtine, felbamate, retigabine, and carisbamate, as it shows remarkable antiseizure efficacy in clinical trials, thus paving the way for the use of this drug to treat patients with focal seizures that have proven to be refractory to other medications [9,10,11,12,13].

Cenobamate mechanism of action has yet to be fully clarified [14]; it was hypothesized, however, that its remarkable clinical efficacy may be the result of a synergistic effect on two different targets: (***i***) inhibition of voltage-gated sodium channels, which involves a a preferential reduction in the persistent (I_NaP_) rather than the transient Na^+^ current (I_NaT_) [15]; (***ii***) positive allosteric modulation at presynaptic and extrasynaptic GABA_A_ receptors, modulating both phasic (I_phasic_) and tonic (I_tonic_) currents and producing an increased inhibitory neurotransmission in the brain independently from the benzodiazepine binding site [16,17,18].

Cenobamate shows 88% of oral bioavailability and is metabolized in the liver through a glucuronide conjugation, catalyzed by uridine 5′-diphospho-glucuronosyltransferase (UGT) 2B7 and, to a minor extent, 2B4, followed by cytochrome P450 (CYP)-mediated oxidation (2E1, 2A6, 2B6, and, to a lesser extent, 2C19 and 3A4/5) [19]. 

Cenobamate, as commonly happens for ASMs, is likely administered in multi-drug therapies. Pharmacokinetic interactions with other ASMs were indeed described, and dose adjustments may be required, in particular for co-administration with lamotrigine (LTG), carbamazepine (CBZ), phenytoin (PHT), phenobarbital (PB) and clobazam (CLB) [9]. Furthermore, as CNB is a relatively new drug in the therapeutic landscape of epilepsy, clinical practice may strongly differ from randomized clinical trials in terms of intra- and inter-individual pharmacokinetic variability. As an example, inter-subjective variability of CNB in humans was described up to 25% for C_max_ and to 35% for AUC from time zero to time t (AUC_t_), while the estimated intra-subject variability was reported to be up to 14% for C_max_ and up to 5% for AUC_t_ [20]. In addition, to limit the risk of drug rash with eosinophilia and systemic symptoms (DRESS), and other serious adverse effects, a ‘‘start low, go slow” titration approach was strongly suggested for CNB, which includes a starting dose of 12.5 mg/day and a 11-week titration phase to reach a target dose of 200 mg/day [11,21]. Finally, although newer ASMs are undoubtedly associated with a better safety and tolerability profile than first-generation drugs, all ASMs are generally associated with a risk of adverse effects [22,23]. Consequently, titration, along with careful dosing and patient monitoring, is an essential component of CNB treatment individualization to mitigate safety concerns related to the use of this drug. In addition, measuring drug plasma concentration in patients undergoing therapeutic treatment with CNB would allow to both define the optimal therapeutic plan for each specific patient and provide useful information on the bioavailability of this drug under different conditions.

To date, only a few methods have been developed and validated to quantify CNB plasma levels in real life. Oh and colleagues published an LC–MS/MS method to describe pharmacokinetics of CNB in rat plasma [24]. During phase 1 and phase 2 studies, human plasma samples were used to validate a LC–MS/MS method. The results of all validation assays were reported in the FDA Center for Drug Evaluation and Research’s Office of Clinical Pharmacology (OCP) section. However, not all analytical procedures and sample preparations were reported in detail, such as mobile phases, chromatographic gradients, organic solvent(s) used for deproteinization of human plasma [25]. A more recent liquid scintillation counting method was validated for investigating CNB pharmacokinetics in healthy male subjects after a single oral dose of [14C]-labeled CNB [26]. 

Here, we describe a rapid UHPLC–MS/MS method that met all criteria requested by international guidelines for bioanalytical method validation. Furthermore, we showed that our UHPLC–MS/MS method was suitable to monitor CNB in human plasma under real-life conditions showing the pharmacokinetics of two patients during their ramp-up dosage. 

## 2. Results

### 2.1. Chromatographic and Mass Spectrometry Parameters Optimization

Cenobamate and the internal standard (IS) were analyzed on a triple-quadrupole mass spectrometer in the positive ionization mode. Available deuterated standards such as felbamate-d4, topiramate-d12 and lamotrigine-13C-d3 were tested (data not shown) as the IS. Lamotrigine -13C-d3 was found to provide better results, both in terms of chromatographic resolution and recovery, and showed good normalization capabilities. To set up an accurate and selective quantification method, the SRM acquisition mode was used. Transitions were chosen by direct injection of pure compounds into the mass spectrometer. CID-induced fragmentation of protonated CNB ([M+H]^+^ m/z 268) produced two major daughter ions, one at m/z 198, produced by the neutral loss of the tetrazole cycle and one at m/z 155 generated by a further elimination of a molecule of isocyanic acid. Since this latter ion was the most intense, the transition 268 → 155 (quantifier) was selected to achieve CNB quantification whereas the transition 268 → 198 (qualifier) was used to confirm the correct identification of the compound. Transition 262 → 217 was used for the IS. To achieve the best chromatographic resolution and reduce the possibility of interferences, a C18 and a pentafluorophenilic (PFP) column were tested. The PFP column showed the best analytical performance. Elution conditions used included a multi-step gradient detailed in the Materials and Methods section. Uniform peaks, with good intensity and short analysis times, were obtained. Retention times of CNB and IS were 1.2 and 1.1 min, respectively (Figure 2).

### 2.2. Method Validation

Tests were performed using plasma samples containing CNB and other ASMs, including levetiracetam (LEV), lamotrigine (LTG), primidone (PRM), oxcarbazepine (OXC), carbamazepine epoxide (CBZ-E), 10-monohydroxy-carbazepine (10-OH-OXC), carbamazepine (CBZ), felbamate (FBM), N-desmethyl-methsuximide (N-DESM), rufinamide (RUF), phenytoin (PHT), methsuximide (MESM), topiramate (TPM), phenobarbital (PB), valproic acid (VPA), ethosuximide (ETS), lacosamide (LCS), zonisamide (ZNS), sulthiame (STM), pregabalin (PGB), brivaracetam (BRV), gabapentin (GBP), stiripentol (STP), vigabatrin (VGB), and tiagabine (TGB). No interfering peaks were observed at either CNB or IS elution times. Figure 3 shows representative chromatograms of blank plasma and samples from patients under CNB treatment (150 mg/die) co-administered with 500 mg/2xdie CBZ and 300 mg/3xdie LCS. 

A calibration curve was built using blank plasma samples spiked with IS only and samples fortified with CNB standard solution to achieve seven calibration points (0, 0.5, 2.5, 5.0, 7.5, 10.0, 15.0 and 20.0 µg/mL). The method showed good linearity over this concentration range (R^2^ = 0.9992). Linear regression of the standard curve was fitted into the equation corresponding to: y = 0.0728x (+/−0.0016) + 0.0115 (+/−0.0027). The adherence to the mathematical model was verified with linearity assumptions and the analysis of residual quantiles. The method proved to be linear in the concentration range selected and the zero hypothesis was rejected. The lower limit of detection (LLOD) and quantification (LLOQ) were evaluated using plasma fortified with CNB, and a 0.02 µg/mL LLOD and a 0.05 µg/mL LLOQ were measured. However, since the expected plasma range in patients was significantly higher, sample containing a CNB concentration corresponding to LLOQ was not included in the calibration curve and in Quality Control samples (QCs). The last point of the calibration curve, set as the upper limit of quantification (ULOQ) of the method, was selected considering the range of expected plasma values. Precision and trueness data (Table 1) were measured through a 5 × 5 experimental model on non-consecutive days. The results obtained were largely satisfying, as most of the values were lower than 10% and never higher than 18%.

Precision and trueness results were evaluated by a one-way ANOVA test, reported in Table 2. F ratios (F_stat_) obtained were smaller than critical F (F_crit_), which was 2.86 (0.05 and 4.20) for both measurements, thus confirming that the biases between repetitions are not statistically significant at the 95% confidence level. 

Plasma samples containing CNB showed good stability when plasma was stored at −20 and −80 °C for over a week. The different storage conditions showed an analyte percentage of degradation higher than 15% when the sample was not frozen and stored for more than 48 hours (h). Stability data are shown in Table 3 and Figure 4. 

Samples obtained from dilutions, showed a mean bias of −5.9% and a mean coefficient of variation (CV) of 3.1%, thus confirming that biological samples dilution does not significantly affect instrumental response.

To evaluate recovery and matrix effect (ME), samples spiked at 1.5, 6.5 and 12.5 µg/mL were used. The average recovery of the plasma extraction was 84 ± 7% for CNB and 84 ± 8% for the IS. Matrix effect was lower than 6% for all samples for both CNB and the IS. These results confirmed that acetonitrile (ACN)-induced protein precipitation was suitable for the proposed application. No carry-over was observed in blank samples after running a high-concentration (30 µg/mL) sample. 

### 2.3. PK Analysis of Patients Undergoing CNB Treatment

As aforementioned, CNB shows a non-linear pharmacokinetic with its systemic exposure which increases more than proportionally with increasing doses. Moreover, the relevant inter-patient variability, also due to potential interactions with other co-administered drugs, highlights the needs of new studies in real-life settings. We investigated the pharmacokinetic profile of CNB in two non-adult patients undergoing CNB ramp-up and exposed to different combinations of ASMs (RUF and VPA for patient 1; CBZ and LCS for patient 2). However, it is important to underline that, at this preliminary stage, we cannot ascertain whether patients were exposed to additional substrates of CYP and UGT enzymes, able of altering the pharmacokinetic profile of CNB. Patients were under treatment for compassionate use at the Child Neuropsychiatry Unit of the University Hospital San Giovanni di Dio e Ruggi d’Aragona of Salerno (Italy). According to the FDA prescription, the dosage was started at 12.5 mg per day and was increased every two weeks until the maintenance dose of 200 mg per day was reached. Plasma levels were monitored for two months. Our data showed a linear increase in CNB plasma concentrations in the dose range of 25–150 mg, while a further increase in the administrated dose produced only a slight increment in plasma drug concentration. Intriguingly, CNB plasma concentrations measured in the two patients at the same administrated doses were sensibly different (Figure 5), although clinically for both subjects a seizure-free state was reported after 45 days of treatment, when they were taking 150 mg per day of CNB. This result confirmed the great inter-individual variability in the bioavailability of CNB and the strong influence that concomitant therapies can have on it. This variability, which may not be easily detected by the patient’s clinical observation alone, requires the use of suitable quantification methods such as the one we have developed.

## 3. Discussion

Therapeutic monitoring of ASMs, which are characterized by a narrow therapeutic window and by several interactions with other drugs, was a consolidated procedure for years [27,28,29,30]. Recently, the requirement to evaluate for these molecules the total and unbound fraction to better define the correlation between concentrations and pharmacological effects has also been highlighted [5,31]. 

Cenobamate is among the newest member of the ASMs family and was characterized by encouraging clinical effects that unfortunately still lack a complete understanding of the underpinning molecular mechanism. The use of CNB in clinical routine undoubtedly needs to be supported by real-life TDM to highlight potential intra- and inter-individual variability among treated patients, to monitor patients compliance and, most importantly, to better define the occurrence of pharmacokinetic interactions with concomitant ASMs, whose influence on CNB metabolism, safety and efficacy has not yet been completely elucidated. 

Our study aimed at developing and validating a novel method for the determination of CNB concentrations in plasma samples. To the best of our knowledge, a limited number of LC–MS/MS methods was described in the literature for the quantification of this drug in human plasma, which explains our effort to set up a new methodology to use in clinical routine. The method described in our study was verified according to the more recent guidelines for the validation of bioanalytical methods [32]. Precision, trueness, reproducibility of the method and stability of the sample at different storage conditions and for different times were investigated. The method has shown good performance in terms of sensitivity, with a short run time and a simple protein precipitation procedure. 

Cenobamate dosage recommendation prescribes a starting dose of 12.5 mg/die, followed by a careful dose titration with a progressive increase every two weeks until the recommended maintenance dose of 200 mg/die is reached. [19]. Thus, it was necessary to develop a method that was linear over a wide range of plasma concentrations, which allowed to monitor the complete titration procedure. Stability tests highlighted that, in our hands, plasma samples can be stored for at least two weeks at −20 or −80 °C. After the extraction procedure, samples can be analyzed up to 48 h, but it is worth noting that samples storage at 4 °C for more than 24 h should be avoided. No analytical interference with other co-administered ASMs with CNB in therapy was observed; this is of utmost importance in the clinical practice since CNB is intended as an adjuvant therapy together with other drugs. 

Our method was used to monitor CNB plasmatic concentrations during a dosing scale-up in two young adult patients. Interestingly, our preliminary data showed that, despite a significant difference in plasma concentrations observed between the two patients, a minimum dose of 50 mg per day resulted in a reduction in seizures for both patients with a seizure-free state reported in both cases at the dose of 150 mg per day. It should be emphasized that many factors may be responsible for this discrepancy in CNB plasma concentrations including a potential inter-individual pharmacodynamic variability, different timing for blood collection, concomitant intake of either ASMs or other drugs, a different Body Mass Index. It is not advisable to advance hypothesis due to the limited number of data available. Nonetheless, our aim at this stage was to optimize and apply to clinical routine a convenient, rapid, selective, and sensitive UHPLC–MS/MS method that proved to be feasible in describing the CNB pharmacokinetic profile of two different patients. Obviously, a much higher number of patients will be necessary to identify the therapeutic range of this novel drug and to better draw the boundaries of CNB inter- and intra-individual variability.

## 4. Materials and Methods

Ultra-pure solvents and formic acid were purchased from Romil (Waterbeach Cambridge, GB); CNB (100 mg) was purchased from DC Chemicals^®^ (Shanghai, China). The deuterated IS lamotrigine-13C3-d3 (1 mg) was obtained from LGC standards (LGC^®^ Group). To prepare QCs and calibration curves, ASMs-free human plasma was obtained from healthy volunteers recruited at the Blood Establishment of the University Hospital “San Giovanni di Dio e Ruggi d’Aragona” in Salerno. Whole-blood samples were collected in BD Vacutainer^®^ tubes containing 2,2′,2″,2‴-(Ethane-1,2-diyldinitrilo) tetra acetic acid (EDTA) as an anticoagulant. Whole blood was centrifuged at 3500× *g* for 6 min, resulting plasma was transferred into clean safe-lock tubes and stored at −20 °C. 

### 4.1. UHPLC–MS/MS Analyses

A Kinetex 2.6 µm PFP 100 Å (2.1 × 50 mm) column was used for chromatographic separation. The mobile phase was composed of solvents A (Water, 0.1% HCOOH) and B (ACN, 0.1% HCOOH). The following multi-step gradient elution was used: 10% B from 0 to 0.3 min, from 10% to 70% B in 0.6 min, then hold at 70% B for 0.3 min, from 70% B to 90% B in 0.3 min, then hold at 90% B for 1 min for washing; from 90% B to 10% B in 0.3 min and then hold at 10% B for 0.7 min for equilibration. Flow rate was set at 0.4 mL/min and column was maintained at 30 °C for the entire run. Five microliters were injected. LC–MS/MS analyses were performed by an Ultimate 3000 UHPLC instrument coupled with an Endura TSQ mass spectrometer, equipped with an electrospray ion source and a triple-quadrupole analyser (Thermo Fisher Scientific, Cambridge, MA, USA). Analyses were performed in the selected reaction monitoring mode (SRM), using the specific transitions for the compound and the IS. Specifically, the positive ionization mode was used and the transition selected to monitor CNB were: m/z 268 ([M+H]^+^) → 155 (quantifier) and m/z 268 → 198 (qualifier); for the IS, m/z 262 ([M+H]^+^) → 217. The collision energy (CE) was 14.8 and 10.2 for CNB quantifier and qualifier, respectively (RF lens 87 for both). The internal standard CE was 27 (RF lens 212). Mass spectrometry parameters were: spray voltage (V) 3500, sheet gas (arb) 40, aux gas (arb) 10, ion transfer tube temperature 350 °C and vaporizer temp 350 °C. Peak areas were measured using the Xcalibur software (Thermo Fisher Scientific). 

### 4.2. Stock Solutions, Calibration Curve and Quality Controls

The CNB stock solution was prepared dissolving the commercially available molecule (solid form) in DMSO at a concentration of 1 mg/mL. Working solutions at a final concentration of 100 µg/mL were obtained by serial dilution of the stock solution using 50% MeOH. The deuterated IS solution was prepared by dissolving neat powder directly in 100% MeOH. Precipitant reagent was obtained by adding the IS to 100% ACN at a final concentration of 2 µg/mL. All solutions were stored at −20 °C until use. An eight-point (0, 0.5, 2.5, 5.0, 7.5, 10.0, 15.0 and 20.0 µg/mL) calibration curve was built by adding CNB in plasma matrix. Quality control samples were prepared at five different concentrations: 0.5 (Lower limit QC = LLQC) 1.5 (low QC = LQC), 6.5 (medium QC = MQC), 12.5 (high QC = HQC) and 20.0 (upper limit QC = ULQC) µg/mL.

### 4.3. Sample Preparation

To obtain analytical samples, 50 µL of calibrators, controls or samples were added to 150 µL of precipitation mixture, consisting of ACN and 2 µg/mL IS. Samples were mixed by vortex and then centrifuged for 10 min at 17,000× *g*. Fifty microliters of clear supernatant was added to an equal volume of water and transferred into conical glass vials. Samples that showed a concentration higher than ULOQ were re-analyzed after a suitable dilution, as described in the dilution integrity test section.

### 4.4. Method Validation

#### 4.4.1. Limits and Linearity

To measure the LLOD (S/N ≥ 3) and the LLOQ (S/N ≥ 5) [32], plasma samples at a CNB concentration of 0.01, 0.02, 0.05 and 0.1 µg/mL were prepared and analyzed in triplicate. Linearity was determined through a calibration curve built using CNB/IS peak area ratios plotted against the corresponding nominal CNB concentrations. Each sample was analyzed in triplicate. The calibration curves were fitted using the linear regression method. Linearity was confirmed by analyzing the goodness-of-fit plot (R square mode) and by residuals analysis.

#### 4.4.2. Inter- and Intra-Day Assay Precision and Trueness

Quality Control samples were used to define precision and trueness. Precision was expressed as %CV and the trueness was evaluated by measuring the relative error, expressed as %bias, between experimental measurements and theoretical concentrations. Mean values within ±15% (±20% for LLOQ) of the nominal concentration were considered acceptable. 

#### 4.4.3. Recovery and Matrix Effects

Recovery and ME were evaluated by analyzing three sets of QCs (LQC, MQC and HQC). Each sample was run in triplicate. Mean recoveries were calculated by comparing the peak areas obtained after extraction of CNB fortified plasma with those prepared by adding the same amounts of drug directly to the extracted blank matrix. Matrix effect, instead, was evaluated by comparing the areas obtained from the same extracted samples, with those obtained by directly adding the same amounts of drug to the blank solvent. Recovery and ME were calculated, respectively, according to the formulae:$$\mathrm{Recovery}=\left[\left(\frac{\sum \mathrm{Extracted}\;\mathrm{samples}\;\mathrm{areas}}{\sum \mathrm{matrix}\;\mathrm{spiked}\;\mathrm{samples}\;\mathrm{area}}\right)*100\right]$$
$$\mathrm{Matrix}\;\mathrm{effect}=\left[\left(\frac{\sum \mathrm{Spiked}\;\mathrm{solvent}\;\mathrm{areas}}{\sum \mathrm{matrix}\;\mathrm{spiked}\;\mathrm{samples}\;\mathrm{area}}\right)*100\right]$$

#### 4.4.4. Interferences and Carry-Over

Possible interferences with other administered drugs were evaluated by injecting samples containing other ASMs, extracted in accordance with the procedure described in this study. The same samples were spiked with known amounts of CNB and re-extracted with the same procedure. All samples were run, analyzed in triplicate and instrumental response was compared. Carry-over was assessed by processing a blank sample after running a sample fortified with CNB at a concentration of 30.0 µg/mL. This was a higher concentration value than the highest calibration point used in our experiments. The percentage of carry-over in the chromatographic run was evaluated.

#### 4.4.5. The Dilution Integrity Test

The dilution integrity test was performed to verify the possibility of quantifying over-range samples. A high-concentration CNB sample (30 µg/mL) was serially diluted to obtain two different final concentrations (5 and 15 µg/mL) falling within the calibration range. Runs were performed in triplicate.

#### 4.4.6. Stability

For clinical purpose, stability tests of CNB in plasma were performed at different times and different storage conditions. Three QCs (LQC, MQC and HQC) were analyzed in triplicate for each different condition. For short-term stability, fortified plasma samples were tested after storage at room temperature for 24 h, at 4 °C, −20 and −80 °C for 24 and 48 h. Medium-long term stability was evaluated at 4 °C, −20 and −80 °C after 7 and 15 days. Stability in the autosampler was tested by storing the extracts at 4 °C for 24 and 48 h. The concentrations obtained from the samples stored under these conditions were compared with the nominal concentrations of the respective QCs prepared at T0.

### 4.5. Statistical Analysis

R software was used to conduct statistical analyses and confirm calibration curves linearity and fitting assumptions. One-way ANOVA tests were applied to evaluate the results; the significance level was α = 0.05 (95% of confidence interval).

### 4.6. Ethics Approval

Human plasma samples used in this study for method validation were provided as waste material by the San Giovanni di Dio e Ruggi d’Aragona University Hospital in Salerno. The clinical data presented concerned patients enrolled at the Clinical Unit of Pediatric Neuropsychiatry of the same hospital, as part of the routine therapeutic monitoring of ASMs. Approval by the ethics committee for compassionate use was obtained for each patient before starting a CNB-based therapy. Patient data were processed in compliance with current legislation on the protection of privacy and according to the procedures of the General Data Protection Regulation (GDPR) n. 2016/67.

## Figures and Tables

**Figure 1 molecules-27-07325-f001:**
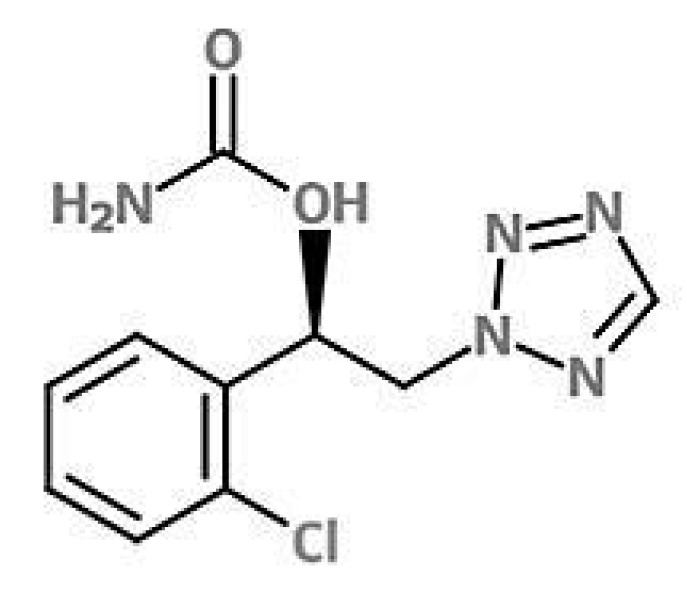
Chemical structure of cenobamate.

**Figure 2 molecules-27-07325-f002:**
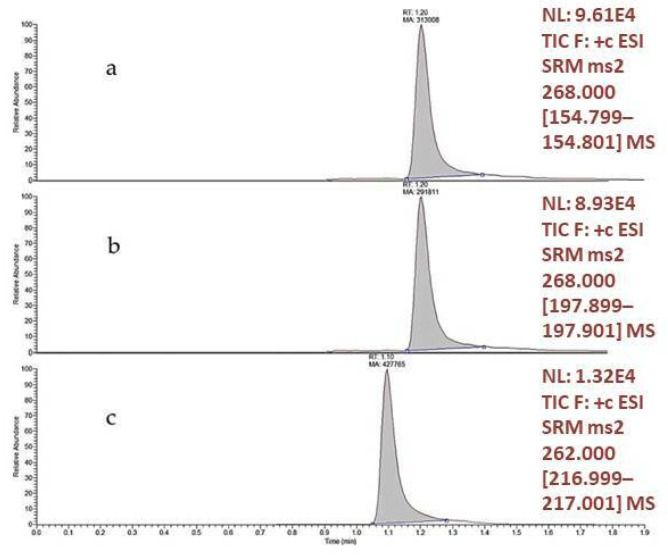
Chromatographic profile of CNB: (**a**) quantifier transition; (**b**) qualifier transition; and the internal standard (**c**).

**Figure 3 molecules-27-07325-f003:**
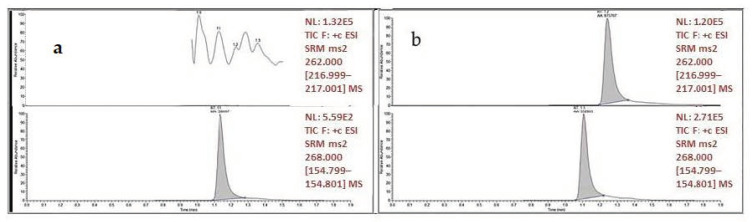
Blank plasma (**a**) vs. patient plasma under CNB treatment and concomitant assumption of other ASMs (**b**). In panel (**a**,**b**), lower chromatograms showing the internal standard peak.

**Figure 4 molecules-27-07325-f004:**
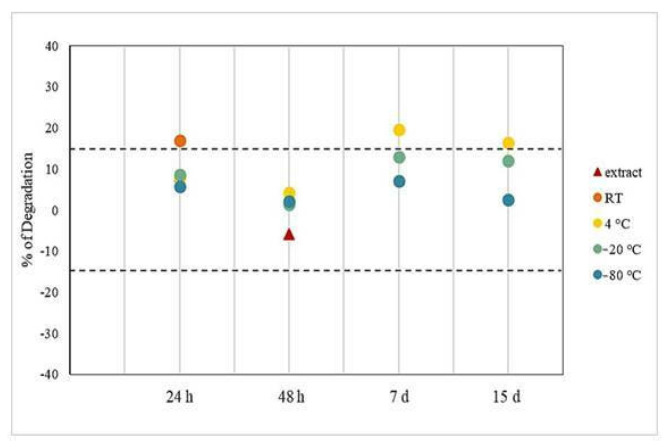
Stability of CNB in extracts and plasma samples at different storage conditions over two weeks, reported as the percentage of molecule degradation. For clarity only the average value reported in Table 3 is here represented. Dashed lines show acceptability range of ±15%.

**Figure 5 molecules-27-07325-f005:**
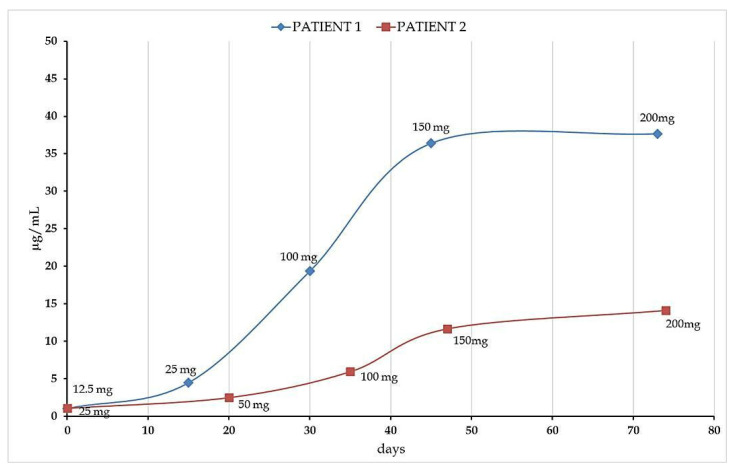
Plasma drug concentrations observed in two young adult patients under CNB treatment. Curve points represent CNB dosage at follow up.

**Table 1 molecules-27-07325-t001:** Inter- and intra-day precision and trueness of our method in human plasma. QCs = quality control samples; LLQC = lower limit QC (0.5 µg/mL); LQC = lower QC (1.5 µg/mL); MQC = medium QC (6.5 µg/mL); HQC = high QC (12.5 µg/mL); ULQC = upper limit QC (20.0 µg/mL); CV = coefficient of variation (see Section 4).

	INTER-DAY									
	Day 1		Day 2		Day 3		Day 4		Day 5	
QCs	%bias	%CV	%bias	%CV	%bias	%CV	%bias	%CV	%bias	%CV
LLQC	−17.8	3.7	−15.0	1.1	−12.7	5.3	−17.0	1.4	4.0	1.2
LQC	0.2	0.3	9.4	3.8	−2.9	1.9	4.6	4.0	−11.3	1.9
MQC	4.7	1.2	11.6	5.0	14.9	3.1	9.7	3.8	−1.3	0.4
HQC	4.6	5.1	0.4	0.3	5.4	0.3	11.0	8.8	11.1	3.8
ULQC	−2.5	0.4	−7.3	0.8	−1.9	3.4	−6.5	3.1	1.0	4.6
	**INTRA-DAY**									
	**Injection 1**		**Injection 2**		**Injection 3**		**Injection 4**		**Injection 5**	
**QCs**	**%bias**	**%CV**	**%bias**	**%CV**	**%bias**	**%CV**	**%bias**	**%CV**	**%bias**	**%CV**
LLQC	−14.8	1.7	−15.7	11.3	−11.4	0.6	−1.6	0.0	−1.3	5.3
LQC	13.0	5.8	0.3	4.5	8.7	0.1	10.9	0.8	12.3	0.1
MQC	9.7	3.8	3.4	0.9	12.6	2.9	12.6	1.0	4.5	0.6
HQC	8.0	7.9	1.8	5.4	11.7	5.6	7.0	0.4	13.3	0.6
ULQC	−12.8	6.8	−16.8	0.9	−8.8	1.3	0.0	5.5	−6.7	2.1

**Table 2 molecules-27-07325-t002:** Results of the one-way ANOVA test for the inter- and intra-day precision and trueness data.

INTER-DAY					
Source of Variation	Degrees of Freedom	Sum of Square	Mean of Square	F Value	Level of Significance
**Between Groups**	4	0.60	0.15	0.002	α = 0.05
**Within Groups**	20	1357.13	67.86		
**Total**	24	1357.73	68.01		
**INTRA-DAY**					
**Source of Variation**	**Degrees of Freedom**	**Sum of Square**	**Mean of Square**	**F Value**	**Level of Significance**
**Between Groups**	4	2.72	0.68	0.011	α = 0.05
**Within Groups**	20	1280.23	64.01		
**Total**	24	1282.95	64.69		

**Table 3 molecules-27-07325-t003:** Percentage of degradation of fortified plasma samples were tested after storage at room temperature for 24 hours (h); at 4 °C, −20 °C and -80 °C for 24 h, 48 h, 7 days (d) and 15 days. Stability in the autosampler was tested by storing the extracts at 4 °C for 24 and 48 h. STDEV = standard deviation (see Section 4).

		Storage Condition			
Storage Time	QCs	RT	4 °C	−20 °C	−80 °C
extract 24 h	LQC	-	9.3	-	-
	MQC	-	11.7	-	-
	HQC	-	4.5	-	-
	Average		8.5		
	STDEV		3.7		
extract 48 h	LQC	-	−10.1	-	-
	MQC	-	−5.4	-	-
	HQC	-	−1.9	-	-
	Average		−5.8		
	STDEV		4.1		
24 h	LQC	14.9	5.1	8.1	4.0
	MQC	21.3	10.2	13.8	9.5
	HQC	14.9	8.3	3.5	3.8
	Average	17.0	7.9	8.5	5.7
	STDEV	3.7	2.6	5.2	3.2
48 h	LQC	-	1.1	3.9	2.1
	MQC	-	7.0	3.6	1.5
	HQC	-	4.6	−3.0	3.1
	Average		4.2	1.5	2.2
	STDEV		3.0	3.9	0.8
7 d	LQC	-	21.9	18.1	9.1
	MQC	-	19.3	13.6	8.7
	HQC	-	17.7	6.9	3.5
	Average		19.6	12.9	7.1
	STDEV		2.1	5.6	3.1
15 d	LQC	-	20.5	12.4	9.6
	MQC	-	14.8	18.0	−2.0
	HQC	-	13.8	5.5	−0.1
	Average		16.4	12.0	7.1
	STDEV		3.6	6.2	6.2

## Data Availability

Not applicable.

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
