# Peer review of "Development and Validation of a UHPLC–MS/MS-Based Method to Quantify Cenobamate in Human Plasma Samples"

_molecules, 2022, doi:10.3390/molecules27217325_

Round 1

Reviewer 1 Report

The article “Development and validation of a UHPLC-MS/MS-based method to quantify cenobamate in human plasma samples.” aims to develop and validate an UHPLC-MS/MS method for the determination of cenomabate (CMB) in plasma and its application to clinical samples. Method development is briefly presented showing very limited data, and most of the data presented is on method validation. However, the structure in which the results and materials and methods are presented makes the article difficult to follow. I would recommend the article be restructured and resubmitted.

Line 89-92: Authors make a detailed description of antiseizure medications (ASMs) and CNB. However, I would suggest reformulating the last part of the introduction by including more data on the detection methods described in the literature to detect CNB in plasma describing their limitations, to finally propose the need of this study and the development and validation of the UHPLC-MS/MS method proposed by the authors.

Line 99: Please define IS

Line 99: “…. in positive ionization mode.”

Line 105-107: Authors must define properly the precursor ions. Do you select the protonated molecule [M+H]+?. To which part of the molecule corresponds the quantifiers and qualifiers ions?

Line 108: Authors state that PFP showed a good performance, albeit CNB and the IS partially coelute. Did you try different gradients of mobile phase to separate these two compounds?

Line 115: Please improve the resolution of Figure 2 and include the description of each ion transition.

Line 121: Please describe the other ASMs that were used.

Line 128: Please improve the resolution of Figure 3 and include the description of each ion transition.

Line 139: “The methods limit of detection and quantitation…”

Materials and method section must be reorganized, there are too many bullet points and some of the sections might be combined.

Lines 264 and 265: Please include the concentration of CNB standard the deuterated internal standard Lamotrigine-13C3-d3.

4.1. Chromatography and 4.2. Mass spectrometry analysis should be described in a single section “UHPLC-MS/MS analyses”. Please describe the gradient of mobile phase without bullet points and include the injection volume. Please describe which are the precursor ions, is m/z 268 the protonated molecule [M+H]+?. Which CE was used for the different ion transitions? What are the ion source and interface parameters (e.g. gas temperature and flows, voltages…)

Line 289: Please change “… -> 15 … ” to “… -> 155 … ”

Reviewer 2 Report

This manuscript contained development of LC-MS/MS determination method of cenobamate and its application. The application data may be clinically useful. However, some questions remained. Please respond to the following questions.   

1. The aim of this study was not clear. Please strengthen the necessity to quantify the concentration in plasma.

2. Figure 2.

1) Peak shapes of cenobamate and IS were clearly asymmetric. How did the authors accurately integrate them?

2) Were the peaks of cenobamate and IS retained on the column?

3. The authors claimed instability of cenobamate in not-frozen plasma matrix. Please discuss how the compound was decomposed.

4. Table 2. Did the results include all tried concentrations?

5. Figure 5.

1) Why were the plasma concentrations deviated between toe patients? Please discuss it.

2) Please discuss relationship between dosage and plasma concentrations.

Reviewer 3 Report

Authors studied the validation of a UHPLC-MS/MS-based method to quantify cenobamate in human plasma samples. Please concern the following comments:

Page 9, Line 287: authors mentioned that’’ Analyses 286 were performed in selected reaction monitoring mode (SRM) while in the line 105, page 3 the mentioned ‘’ To this purpose, MRM mode was used and the transitions selected’’. Please clarify.

Stability results should be mentioned in details in the form of table.

The gradient steps of the elution should be clearly mentioned (as it mentioned in the inlet file of the method)

4.2. The expression of powder is not correct, please change.

4.4. vortexed is not correct, please change.

Round 2

Reviewer 1 Report

The authors have modified the article according to the suggestions I proposed improving the quality of the manuscript.

Reviewer 2 Report

This revised manuscript was well revised; therefore, it should be accepted as is.

Reviewer 3 Report

The authors responded and clarified all the points in my comment